# META-LEARNING DEEP ENERGY-BASED MEMORY MODELS

**Sergey Bartunov**
DeepMind
London, United Kingdom
`bartunov@google.com`

**Jack W Rae**
DeepMind
London, United Kingdom
`jwrae@google.com`

**Simon Osindero**
DeepMind
London, United Kingdom
`osindero@google.com`

**Timothy P Lillicrap**
DeepMind
London, United Kingdom
`countzero@google.com`

## ABSTRACT

We study the problem of learning an associative memory model – a system which is able to retrieve a remembered pattern based on its distorted or incomplete version. Attractor networks provide a sound model of associative memory: patterns are stored as attractors of the network dynamics and associative retrieval is performed by running the dynamics starting from a query pattern until it converges to an attractor. In such models the dynamics are often implemented as an optimization procedure that minimizes an energy function, such as in the classical Hopfield network. In general it is difficult to derive a writing rule for a given dynamics and energy that is both compressive and fast. Thus, most research in energy-based memory has been limited either to tractable energy models not expressive enough to handle complex high-dimensional objects such as natural images, or to models that do not offer fast writing. We present a novel meta-learning approach to energy-based memory models (EBMM) that allows one to use an arbitrary neural architecture as an energy model and quickly store patterns in its weights. We demonstrate experimentally that our EBMM approach can build compressed memories for synthetic and natural data, and is capable of associative retrieval that outperforms existing memory systems in terms of the reconstruction error and compression rate.

## 1 INTRODUCTION

Associative memory has long been of interest to neuroscience and machine learning communities (Willshaw et al., 1969; Hopfield, 1982; Kanerva, 1988). This interest has generated many proposals for associative memory models, both biological and synthetic. These models address the problem of storing a set of patterns in such a way that a stored pattern can be retrieved based on a partially known or distorted version. This kind of retrieval from memory is known as auto-association.

Due to the generality of associative retrieval, successful implementations of associative memory models have the potential to impact many applications. Attractor networks provide one well-grounded foundation for associative memory models (Amit & Amit, 1992). Patterns are stored in such a way that they become attractors of the update dynamics defined by the network. Then, if a query pattern that preserves sufficient information for association lies in the basin of attraction for the original stored pattern, a trajectory initialized by the query will converge to the stored pattern.

A variety of implementations of the general attractor principle have been proposed. The classical Hopfield network (Hopfield, 1982), for example, defines a simple quadratic energy function whose parameters serve as a memory. The update dynamics in Hopfield networks iteratively minimize the energy by changing elements of the pattern until it converges to a minimum, typically corresponding to one of the stored patterns. The goal of the writing process is to find parameter values such that

Figure 1: A schematic illustration of EBMM. The energy function is modelled by a neural network. The writing rule is then implemented as a weight update, producing parameters $\theta$ from the initialization $\bar{\theta}$, such that the stored patterns $x_1$, $x_2$, $x_3$ become local minima of the energy (see Section 3). Local minima are attractors for gradient descent which implements associative retrieval starting from a query $\tilde{x}$, in this case a distorted version of $x_3$.

the stored patterns become attractors for the optimization process and such that, ideally, no spurious attractors are created.

Many different learning rules have been proposed for Hopfield energy models, and the simplicity of the model affords compelling closed-form analysis (Storkey & Valabregue, 1999). At the same time, Hopfield memory models have fundamental limitations: (1) It is not possible to add capacity for more stored patterns by increasing the number of parameters since the number of parameters in a Hopfield network is quadratic in the dimensionality of the patterns. (2) The model lacks a means of modelling the higher-order dependencies that exist in real-world data.

In domains such as natural images, the potentially large dimensionality of an input makes it both ineffective and often unnecessary to model global dependencies among raw input measurements. In fact, many auto-correlations that exist in real-world perceptual data can be efficiently compressed without significant sacrifice of fidelity using either algorithmic (Wallace, 1992; Candes & Tao, 2004) or machine learning tools (Gregor et al., 2016; Toderici et al., 2017). The success of existing deep learning techniques suggests a more efficient recipe for processing high-dimensional inputs by modelling a hierarchy of signals with restricted or local dependencies (LeCun et al., 1995). In this paper we use a similar idea for building an associative memory: *use a deep network's weights to store and retrieve data*.

**Fast writing rules**  A variety of energy-based memory models have been proposed since the original Hopfield network to mitigate its limitations (Hinton et al., 2006b; Du & Mordatch, 2019). Restricted Boltzmann Machines (RBMs) (Hinton, 2012) add capacity to the model by introducing latent variables, and deep variants of RBMs (Hinton et al., 2006b; Salakhutdinov & Larochelle, 2010) afford more expressive energy functions. Unfortunately, training Boltzmann machines remains challenging, and while recent probabilistic models such as variational auto-encoders (Kingma & Welling, 2013; Rezende et al., 2014) are easier to train, they nevertheless pay the price for expressivity in the form of slow writing. While Hopfield networks memorize patterns quickly using a simple Hebbian rule, deep probabilistic models are *slow* in that they rely on gradient training that requires many updates (typically thousands or more) to settle new inputs into the weights of a network. Hence, writing memories via parametric gradient based optimization is not straightforwardly applicable to memory problems where fast adaptation is a crucial requirement. In contrast, and by explicit design, our proposed method enjoys *fast writing*, requiring few parameter updates (we employ just 5 steps) to write new inputs into the weights of the net once meta-learning is complete. It also enjoys *fast reading*, requiring few gradient descent steps (again just 5 in our experiments) to retrieve a pattern. Furthermore, our writing rules are also fast in the sense that they use $O(N)$ operations to store $N$ patterns in the memory – this scaling is the best one can hope for without additional assumptions.

We propose a novel approach that leverages meta-learning to enable fast storage of patterns into the weights of arbitrarily structured neural networks, as well as fast associative retrieval. Our networks output a single scalar value which we treat as an energy function whose parameters implement a distributed storage scheme. We use gradient-based reading dynamics and meta-learn a writing rule in the form of truncated gradient descent over the parameters defining the energy function. We show that the proposed approach enables compression via efficient utilization of network weights, as well as fast-converging attractor dynamics.

## 2    RETRIEVAL IN ENERGY-BASED MODELS

We focus on attractor networks as a basis for associative memory. Attractor networks define update dynamics for iterative evolution of the input pattern: $\mathbf{x}^{(k+1)} = f(\mathbf{x}^{(k)})$.

For simplicity, we will assume that this process is discrete in time and deterministic, however there are examples of both continuous-time (Yoon et al., 2013) and stochastic dynamics (Aarts & Korst, 1988). A fixed-point attractor of deterministic dynamics can be defined as a point $\mathbf{x}$ for which it converges, i.e. $\mathbf{x} = f(\mathbf{x})$. Learning the associative memory in the attractor network is then equivalent to learning the dynamics $f$ such that its fixed-point attractors are the stored patterns and the corresponding basins of attraction are sufficiently wide for retrieval.

An energy-based attractor network is defined by the energy function $E(\mathbf{x})$ mapping an input object $\mathbf{x} \in \mathcal{X}$ to a real scalar value. A particular model may then impose additional requirements on the energy function. For example if the model has a probabilistic interpretation, the energy function is usually a negative unnormalized logarithm of the object probability $\log p(\mathbf{x}) = -E(\mathbf{x}) + \text{const}$, implying that the energy has to be well-behaved for the normalizing constant to exist. In our case no such constraints are put on the energy.

The attractor dynamics in energy-based models is often implemented either by iterative energy optimization (Hopfield, 1982) or sampling (Aarts & Korst, 1988). In the optimization case considered further in the paper, attractors are conveniently defined as local minimizers of the energy function.

While a particular energy function may suggest a number of different optimization schemes for retrieval, convergence to a local minimum of an arbitrary function is NP-hard. Thus, we consider a class of energy functions that are differentiable on $\mathcal{X} \subseteq \mathbb{R}^d$, bounded from below and define the update dynamics over $k = 1, \ldots, K$ steps via gradient descent:

$$\mathbf{read}(\tilde{\mathbf{x}}; \boldsymbol{\theta}) = \mathbf{x}^{(K)}, \quad \mathbf{x}^{(k+1)} = \mathbf{x}^{(k)} - \gamma^{(k)} \nabla_{\mathbf{x}} E(\mathbf{x}^{(k)}), \quad \mathbf{x}^{(0)} = \tilde{\mathbf{x}}. \tag{1}$$

With appropriately set step sizes $\{\gamma^{(k)}\}_{k=0}^{K}$ this procedure asymptotically converges to a local minimum of energy $E(\mathbf{x})$ (Nesterov, 2013). Since asymptotic convergence may be not enough for practical applications, we truncate the optimization procedure (1) at $K$ steps and treat $\mathbf{x}^{(K)}$ as a result of the retrieval. While vanilla gradient descent (1) is sufficient to implement retrieval, in our experiments we employ a number of extensions, such as the use of Nesterov momentum and projected gradients, which are thoroughly described in Appendix B.

Relying on the generic optimization procedure allows us to translate the problem of designing update dynamics with desirable properties to constructing an appropriate energy function, which in general is equally difficult. In the next section we discuss how to tackle this difficulty.

## 3    META-LEARNING GRADIENT-BASED WRITING RULES

As discussed in previous sections, our ambition is to be able to use any scalar-output neural network as an energy function for associate retrieval. We assume a parametric model $E(\mathbf{x}; \boldsymbol{\theta})$ differentiable in both $\mathbf{x}$ and $\boldsymbol{\theta}$, and bounded from below as a function of $\mathbf{x}$. These are mild assumptions that are often met in the existing neural architectures with an appropriate choice of activation functions, e.g. tanh.

The writing rule then compresses input patterns $X = \{\mathbf{x}_1, \mathbf{x}_2, \ldots, \mathbf{x}_N\}$ into parameters $\boldsymbol{\theta}$ such that each of the stored patterns becomes a local minimum of $E(\mathbf{x}; \boldsymbol{\theta})$ or, equivalently, creates a basin of attraction for gradient descent in the pattern space.

This property can be practically quantified by the reconstruction error, e.g. mean squared error, between the stored pattern $\mathbf{x}$ and the pattern $\mathbf{read}(\tilde{\mathbf{x}}; \boldsymbol{\theta})$ retrieved from a distorted version of $\mathbf{x}$:

$$\mathcal{L}(X, \boldsymbol{\theta}) = \frac{1}{N} \sum_{i=1}^{N} \mathbb{E}_{p(\tilde{\mathbf{x}}_i | \mathbf{x}_i)} \left[ ||\mathbf{x}_i - \mathbf{read}(\tilde{\mathbf{x}}_i; \boldsymbol{\theta})||_2^2 \right]. \tag{2}$$

Here we assume a known, potentially stochastic distortion model $p(\tilde{\mathbf{x}}|\mathbf{x})$ such as randomly erasing certain number of dimensions, or salt-and-pepper noise. While one can consider loss (2) as a function of network parameters $\boldsymbol{\theta}$ and call minimization of this loss with a conventional optimization method

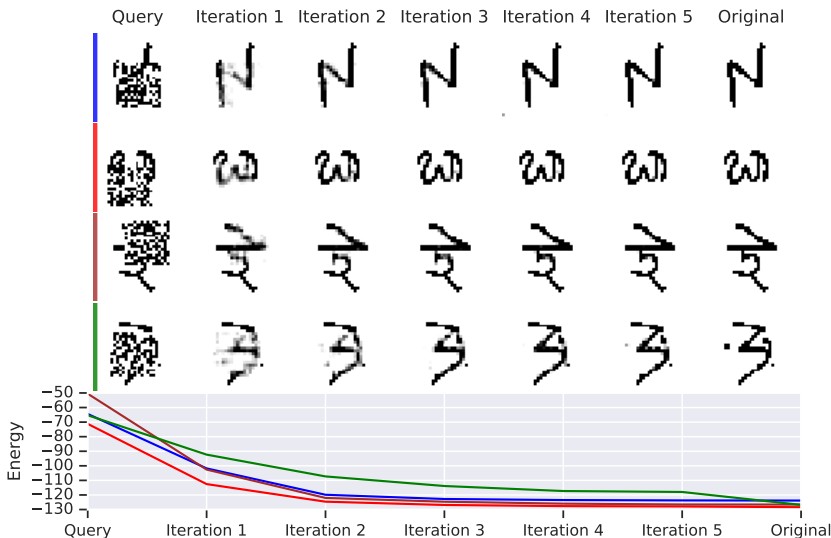

Figure 2: Visualization of gradient descent iterations during retrieval of Omniglot characters (largest model). Four random images are shown from the batch of 64.

a writing rule — it will require many optimization steps to obtain a satisfactory solution and thus does not fall into our definition of fast writing rules (Santoro et al., 2016).

Hence, we explore a different approach to designing a fast writing rule inspired by recently proposed gradient-based meta-learning techniques (Finn et al., 2017) which we call meta-learning energy-based memory models (EBMM). Namely we perform many write and read optimization procedures with a small number of iterations for several sets of write and read observations, and backpropagate into the initial parameters of $\theta$ — to learn a good starting location for fast optimization. As usual, we assume that we have access to the underlying data distribution $p_d(X)$ over data batches of interest $X$ from which we can sample sufficiently many training datasets, even if the actual batch our memory model will be used to store (at test time) is not available at the training time (Santoro et al., 2016).

The straightforward application of gradient-based meta-learning to the loss (2) is problematic, because we generally cannot evaluate or differentiate through the expectation over stochasticity of the distortion model in a way that is reliable enough for adaptation, because as the dimensionality of the pattern space grows the number of possible (and representative) distortions grows exponentially.

Instead, we define a different *writing loss* $\mathcal{W}$, minimizing which serves as a proxy for ensuring that input patterns are local minima for the energy $E(\mathbf{x}; \boldsymbol{\theta})$, but does not require costly retrieval of exponential number of distorted queries.

$$\mathcal{W}(\mathbf{x}, \boldsymbol{\theta}) = E(\mathbf{x}; \boldsymbol{\theta}) + \alpha ||\nabla_x E(\mathbf{x}; \boldsymbol{\theta})||_2^2 + \beta ||\boldsymbol{\theta} - \bar{\boldsymbol{\theta}}||_2^2. \tag{3}$$

As one can see, the writing loss (3) consists of three terms. The first term is simply the energy value which we would like to be small for stored patterns relative to non-stored patterns. The condition for $\mathbf{x}$ to be a local minimum of $E(\mathbf{x}; \boldsymbol{\theta})$ is two-fold: first, the gradient at $\mathbf{x}$ is zero, which is captured by the second term of the writing loss, and, second, the hessian is positive-definite. The latter condition is difficult to express in a form that admits efficient optimization and we found that meta-learning using just first two terms in the writing loss is sufficient. Finally, the third term limits deviation from initial or prior parameters $\bar{\boldsymbol{\theta}}$ which we found helpful from optimization perspective (see Appendix D for more details).

We use truncated gradient descent on the writing loss (3) to implement the writing rule:

$$\mathbf{write}(X) = \boldsymbol{\theta}^{(T)}, \quad \boldsymbol{\theta}^{(t+1)} = \boldsymbol{\theta}^{(t)} - \eta^{(t)} \frac{1}{N} \sum_{i=1}^{N} \nabla_{\boldsymbol{\theta}} \mathcal{W}(\mathbf{x}_i, \boldsymbol{\theta}^{(t)}), \quad \boldsymbol{\theta}^{(0)} = \bar{\boldsymbol{\theta}} \tag{4}$$

To ensure that gradient updates (4) are useful for minimization of the reconstruction error (2) we train the combination of retrieval and writing rules end-to-end, meta-learning initial parameters $\bar{\boldsymbol{\theta}}$,

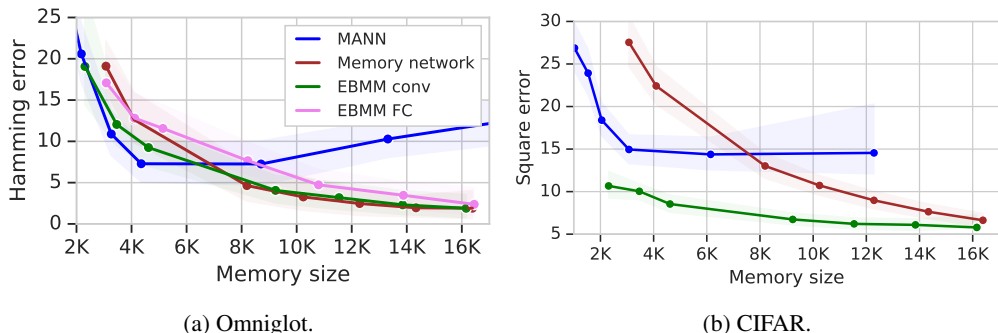

(a) Omniglot.

(b) CIFAR.

Figure 3: Distortion (reconstruction error) vs rate (memory size) analysis on batches of 64 images.

learning rate schedules $\mathbf{r} = (\{\gamma^{(k)}\}_{k=1}^{K}, \{\eta^{(t)}\}_{t=1}^{T})$ and meta-parameters $\tau = (\alpha, \beta)$ to perform well on random sets of patterns from the batch distribution $p_d(X)$:

$$\text{minimize } \mathbb{E}_{X \sim p_d(X)} \left[ \mathcal{L}(X, \mathbf{write}(X)) \right] \text{ for } \bar{\theta}, \mathbf{r}, \tau. \tag{5}$$

In our experiments, $p(X)$ is simply a distribution over batches of certain size $N$ sampled uniformly from the training (or testing - during evaluation) set. Parameters $\theta = \mathbf{write}(X)$ are produced by storing $X$ in the memory using the writing procedure (4). Once stored, distorted versions of $X$ can be retrieved and we can evaluate and optimize the reconstruction error (2).

Crucially, the proposed EBMM implements both $\mathbf{read}(\mathbf{x}, \theta)$ and $\mathbf{write}(X)$ operations via truncated gradient descent which can be itself differentiated through in order to set up a tractable meta-learning problem. While truncated gradient descent is not guaranteed to converge, reading and writing rules are trained jointly to minimize the reconstruction error (2) and thus ensure that they converge *sufficiently* fast. This property turns this potential drawback of the method to its advantage over provably convergent, but slow models. It also relaxes the necessity of stored patterns to create too well-behaved basins of attraction because if, for example, a stored pattern creates a nuisance attractor in the dangerous proximity of the main one, the gradient descent (1) might successfully pass it with appropriately learned step sizes $\gamma$.

## 4 EXPERIMENTS

In this section we experimentally evaluate EBMM on a number of real-world image datasets. The performance of EBMM is compared to a set of relevant baselines: Long-Short Term Memory (LSTM) (Hochreiter & Schmidhuber, 1997), the classical Hopfield network (Hopfield, 1982), Memory-Augmented Neural Networks (MANN) (Santoro et al., 2016) (which are a variant of the Differentiable Neural Computer (Graves et al., 2016)), Memory Networks (Weston et al., 2014), Differentiable Plasticity model of Miconi et al. (2018) (a generalization of the Fast-weights RNN (Ba et al., 2016)) and Dynamic Kanerva Machine (Wu et al., 2018). Some of these baselines failed to learn at all for real-world images. In the Appendix A.2 we provide additional experiments with random binary strings with a larger set of representative models.

The experimental procedure is the following: we write a fixed-sized batch of images into a memory model, then corrupt a random block of the written image to form a query and let the model retrieve the originally stored image. By varying the memory size and repeating this procedure, we perform distortion/rate analysis, i.e. we measure how well a memory model can retrieve a remembered pattern for a given memory size. Meta-learning have been performed on the canonical train splits of each dataset and testing on the test splits. Batches were sampled uniformly, see Appendix A.7 for the performance study on correlated batches.

We define memory size as a number of `float32` numbers used to represent a modifiable part of the model. In the case of EBMM it is the subset of all network weights that are modified by the gradient descent (4), for other models it is size of the state, e.g. the number of slots $\times$ the slot size for a Memory Network. To ensure fair comparison, all models use the same encoder (and decoder, when applicable) networks, which architectures are described in Appendix C. In all experiments EBMM used $K = 5$ read iterations and $T = 5$ write iterations.

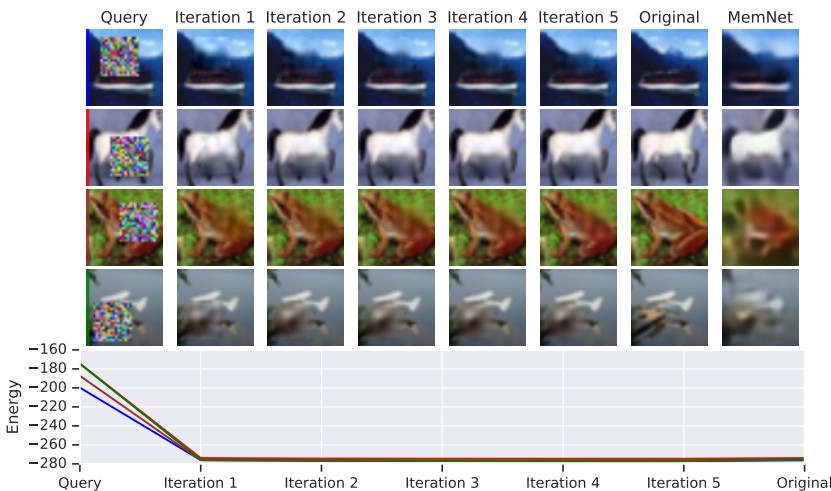

Figure 4: Visualization of gradient descent iterations during retrieval of CIFAR images. The last column contains reconstructions from Memory networks (both models use 10k memory).

### 4.1 OMNIGLOT CHARACTERS

We begin with experiments on the Omniglot dataset (Lake et al., 2015) which is now a standard evaluation of fast adaptation models. For simplicity of comparison with other models, we downscaled the images to $32 \times 32$ size and binarized them using a $0.5$ threshold. We use Hamming distance as the evaluation metric. For training and evaluation we apply a $16 \times 16$ randomly positioned binary distortions (see Figure 2 for example).

We explored two versions of EBMM for this experiment that use parts of fully-connected (FC, see Appendix C.2) and convolutional (conv, Appendix C.3) layers in a 3-block ResNet (He et al., 2016) as writable memory.

Figure 3a contains the distortion-rate analysis of different models which in this case is the Hamming distance as a function of memory size. We can see that there are two modes in the model behaviour. For small memory sizes, learning a lossless storage becomes a hard problem and all models have to find an efficient compression strategy, where most of the difference between models can be observed. However, after a certain critical memory size it becomes possible to rely just on the autoencoding which in the case of a relatively simple dataset such as Omniglot can be efficiently tackled by the ResNet architecture we are using. Hence, even Memory Networks that do not employ any compression mechanisms beyond using distributed representations can retrieve original images almost perfectly. In this experiment MANN has been able to learn the most efficient compression strategy, but could not make use of larger memory. EBMM performed well both in the high and low compression regimes with convolutional memory being more efficient over the fully-connected memory. Further, in CIFAR and ImageNet experiments we only use the convolutional version of EBMM.

We visualize the process of associative retrieval in Figure 2. The model successfully detected distorted parts of images and clearly managed to retrieve the original pixel intensities. We also show energy levels of the distorted query image, the recalled images through 5 read iterations, and the original image. In most cases we found the energy of the retrieved images matching energy of the originals, however, an error would occur when they sometimes do not match (see the green example).

### 4.2 REAL IMAGES FROM CIFAR-10

We conducted a similiar study on the CIFAR dataset. Here we used the same network architecture as in the Omniglot experiment. The only difference in the experimental setup is that we used squared error as an evaluation metric since the data is continuous RGB images.

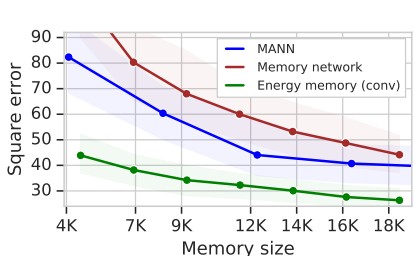



(a) Distortion-rate analysis on ImageNet. (b) Retrieval of 64x64 ImageNet images (all models have ≈18K memory).

Figure 5: ImageNet results.

Figure 3b contains the corresponding distortion-rate analysis. EBMM clearly dominates in the comparison. One important reason for that is the ability of the model to detect the distorted part of the image so it can avoid paying the reconstruction loss for the rest of the image. Moreover, unlike Omniglot where images can be almost perfectly reconstructed by an autoencoder with a large enough code, CIFAR images have much more variety and larger channel depth. This makes an efficient *joint* storage of a batch as important as an ability to provide a good decoding of the stored original.

Gradient descent iterations shown in Figure 4 demonstrate the successful application of the model to natural images. Due to the higher complexity of the dataset, the reconstructions are imperfect, however the original patterns are clearly recognizable. Interestingly, the learned optimization schedule starts with one big gradient step providing a coarse guess that is then gradually refined.

### 4.3 IMAGENET 64X64

We further investigate the ability of EBMM to handle complex visual datasets by applying the model to $64 \times 64$ ImageNet. Similarly to the CIFAR experiment, we construct queries by corrupting a quarter of the image with $32 \times 32$ random masks. The model is based on a 4-block version of the CIFAR network. While the network itself is rather modest compared to existing ImageNet classifiers, the sequential training regime resembling large-state recurrent networks prevents us from using anything significantly bigger than a CIFAR model. Due to prohibitively expensive computations required by experimenting at this scale, we also had to decrease the batch size to 32.

The distortion-rate analysis (Figure 5a) shows the behaviour similar to the CIFAR experiment. EBMM pays less reconstruction error than other models and MANN demonstrates better performance than Memory Networks for smaller memory sizes; however, the asymptotic behaviour of these two models will likely match.

The qualitative results are shown in Figure 5b. Despite the arguably more difficult images, EBMM is able to capture shape and color information, although not in high detail. We believe this could likely be mitigated by using larger models. Additionally, using techniques such as perceptual losses (Johnson et al., 2016) instead of naive pixel-wise reconstruction errors can improve visual quality with the existing architectures, but we leave these ideas for future work.

### 4.4 ANALYSIS OF ENERGY LEVELS

We were also interested in whether energy values provided by EBMM are interpretable and whether they can be used for associative retrieval. We took an Omniglot model and inspected energy levels of different types of patterns. It appears that, despite not being explicitly trained to, EBMM in many cases could discriminate between in-memory and out-of-memory patterns, see Figure 6. Moreover, distorted patterns had even higher energy than simply unknown patterns. Out-of-distribution patterns, here modelled as binarized CIFAR images, can be seen as clear outliers.

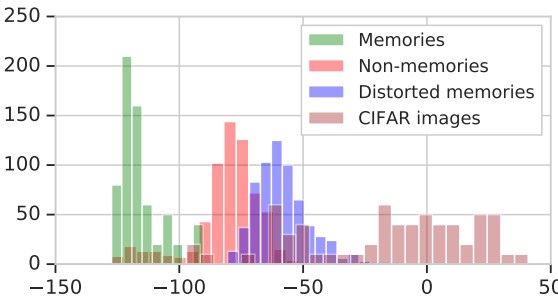

Figure 6: Energy distributions of different classes of patterns under an Omniglot model. *Memories* are the patterns written into memory, *non-memories* are other randomly sampled images and *distorted memories* are the written patterns distorted as during the retrieval. *CIFAR* images were produced by binarizing the original RGB images and serve as out-of-distribution samples.

## 5 RELATED WORK

Deep neural networks are capable of both compression (Parkhi et al., 2015; Kraska et al., 2018), and memorizing training patterns (Zhang et al., 2016). Taken together, these properties make deep networks an attractive candidate for memory models, with both exact recall and compressive capabilities. However, there exists a natural trade-off between the speed of writing and the realizable capacity of a model (Ba et al., 2016). Approaches similar to ours in their use of gradient descent dynamics, but lacking fast writing, have been proposed by Hinton et al. (2006a) and recently revisited by Xie et al. (2016); Nijkamp et al. (2019); Du & Mordatch (2019). Krotov & Hopfield (2016) also extended the classical Hopfield network to a larger family of non-quadratic energy functions with more capacity. In general it is difficult to derive a writing rule for a given dynamics equation or an energy model which we attempt to address in this work.

The idea of meta-learning (Thrun & Pratt, 2012; Hochreiter et al., 2001) has found many successful applications in few-shot supervised (Santoro et al., 2016; Vinyals et al., 2016) and unsupervised learning (Bartunov & Vetrov, 2016; Reed et al., 2017). Our model is particularly influenced by works of Andrychowicz et al. (2016) and Finn et al. (2017), which experiment with meta-learning efficient optimization schedules and, perhaps, can be seen as an ultimate instance of this principle since we implement both learning and inference procedures as optimization. Perhaps the most prominent existing application of meta-learning for associative retrieval is found in the Kanerva Machine (Wu et al., 2018), which combines a variational auto-encoder with a latent linear model to serve as an addressable memory. The Kanerva machine benefits from a high-level representation extracted by the auto-encoder. However, its linear model can only represent convex combinations of memory slots and is thus less expressive than distributed storage realizable in weights of a deep network.

We described literature on associative and energy-based memory in Section 1, but other types of memory should be mentioned in connection with our work. Many recurrent architectures aim at maintaining efficient compressive memory (Graves et al., 2016; Rae et al., 2018). Models developed by Ba et al. (2016) and Miconi et al. (2018) enable associative recall by combining standard RNNs with structures similar to Hopfield network. And, recently Munkhdalai et al. (2019) explored the idea of using arbitrary feed-forward networks as a key-value storage.

Finally, the idea of learning a surrogate model to define a gradient field useful for a problem of interest has a number of incarnations. Putzky & Welling (2017) jointly learn an energy model and an optimizer to perform denoising or impainting of images. Marino et al. (2018) use gradient descent on an energy defined by variational lower bound for improving variational approximations. And, Belanger et al. (2017) formulate a generic framework for energy-based prediction driven by gradient descent dynamics. A detailed explanation of the learning through optimization with applications in control can be found in (Amos, 2019).

Modern deep learning made a departure from the earlier works on energy-based models such as Boltzmann machines and approach image manipulation tasks using techniques such as aforementioned variational autoencoders or generative adversarial networks (GANs) (Goodfellow et al., 2014). While

these models indeed constitute state of the art for learning powerful *prior* models of data that can perform some kind of associative retrieval, they naturally lack fast memory capabilities. In contrast, the approach proposed in this work addresses the problem of jointly learning a strong prior model and an efficient memory which can be used in combination with these techniques. For example, one can replace the plain reconstruction error with a perceptual loss incurred by a GAN discriminator or use EBMM to store representations extracted by a VAE.

While both VAEs and GANs can be also be equipped with memory (as done e.g. by Wu et al. (2018)), energy-based formulation allows us to employ arbitrary neural network parameters as an associative storage and make use of generality of gradient-based meta-learning. As we show in the additional experiments on binary strings in Appendix A.2, EBMM is applicable not only to high-dimensional natural data, but also to uniformly generated binary strings where no prior can be useful. At the same time, evaluation of non-memory baselines in Appendix A.3 demonstrates that the combination of a good prior and memory as in EBMM achieves significantly better performance than just a prior, even suited with a much larger network.

## 6 CONCLUSION AND FUTURE WORK

We introduced a novel learning method for deep associative memory systems. Our method benefits from the recent progress in deep learning so that we can use a very large class of neural networks both for learning representations and for storing patterns in network weights. At the same time, we are not bound by slow gradient learning thanks to meta-learning of fast writing rules. We showed that our method is applicable in a variety of domains from non-compressible (binary strings; see Appendix) to highly compressible (natural images) and that the resulting memory system uses available capacity efficiently. We believe that more elaborate architecture search could lead to stronger results on par with state-of-the-art generative models.

The existing limitation of EBMM is the batch writing assumption, which is in principle possible to relax. This would enable embedding of the model in reinforcement learning agents or into other tasks requiring online-updating memory. Employing more significantly more optimization steps does not seem to be necessary at the moment, however, scaling up to larger batches or sequences of patterns will face the bottleneck of recurrent training. Implicit differentiation techniques (Liao et al., 2018), reversible learning (Maclaurin et al., 2015) or synthetic gradients (Jaderberg et al., 2017) may be promising directions towards overcoming this limitation. It would be also interesting to explore a stochastic variant of EBMM that could return different associations in the presence of uncertainty caused by compression. Finally, many general principles of learning attractor models with desired properties are yet to be discovered and we believe that our results provide a good motivation for this line of research.

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

Table 1: Number of error bits in retrieved binary patterns.

| # PATTERNS / METHOD | 16 | 32 | 48 | 64 | 96 |
|---|---|---|---|---|---|
| HOPFIELD NETWORK, HEBB RULE | 0.4 | 5.0 | 9.8 | 13.0 | 16.5 |
| HOPFIELD NETWORK, STORKEY RULE | 0.0 | 0.9 | 6.3 | 11.3 | 17.1 |
| HOPFIELD NETWORK, PSEUDO-INVERSE RULE | 0.0 | 0.0 | 0.3 | 4.3 | 22.5 |
| DIFFERENTIABLE PLASTICITY (MICONI ET AL., 2018) | 3.0 | 13.2 | 20.8 | 26.3 | 34.9 |
| MANN (SANTORO ET AL., 2016) | 0.1 | 0.2 | 1.8 | 4.25 | 9.6 |
| LSTM (HOCHREITER & SCHMIDHUBER, 1997) | 30 | 58 | 63 | 64 | 64 |
| MEMORY NETWORKS (WESTON ET AL., 2014) | **0.0** | **0.0** | **0.0** | **0.0** | 10.5 |
| **EBMM RNN** | **0.0** | **0.0** | **0.1** | 0.5 | **4.2** |

## A  ADDITIONAL EXPERIMENTAL DETAILS

We train all models using AdamW optimizer (Loshchilov & Hutter, 2017) with learning rate $5 \times 10^{-5}$ and weight decay $10^{-6}$, all other parameters set to AdamW defaults. We also apply gradient clipping by global norm at $0.05$. All models were allowed to train for $2 \times 10^6$ gradient updates or 1 week whichever ended first. All baseline models always made more updates than EBMM.

One instance of each model has been trained. Error bars showed on the figures correspond to 5- and 95-percentiles computed on a 1000 of random batches.

In all experiments we used initialization scheme proposed by He et al. (2015).

### A.1  FAILURE MODES OF BASELINE MODELS

Image retrieval appeared to be difficult for a number of baselines.

LSTM failed to train due to quadratic growth of the hidden-to-hidden weight matrix with increase of the hidden state size. Even moderately large hidden states were prohibitive for training on a modern GPU.

Differential plasticity additionally struggled to train when using a deep representation instead of the raw image data. We hypothesize that it was challenging for the encoder-decoder pair to train simultaneously with the recurrent memory, because in the binary experiment, while not performing the best, the model managed to learn a memorization strategy.

Finally, the Kanerva machine could not handle the relatively strong noise we used in this task. By design, Kanerva machine is agnostic to the noise model and is trained simply to maximize the data likelihood, without meta-learning a particular de-noising scheme. In the presence of the strong noise it failed to train on sequences longer than 4 images.

### A.2  EXPERIMENTS WITH RANDOM BINARY PATTERNS

Besides highly-structured patterns such as Omniglot or ImageNet images we also conducted experiments on random binary patterns – the classical setting in which associative memory models have been evaluated. While such random patterns are not compressible in expectation due to lack of any internal structure, by this experiment we examine the efficiency of a learned coding scheme, i.e. how well can each of the models store binary information in the floating point format.

We generate random 128-dimensional patterns, each dimension of which takes values of $-1$ or $+1$ with equal probability, corrupt half of the bits and use this as a query for associative retrieval. We compare EBMM employing a simple fully recurrent network (an RNN using the same input at each iteration, see Appendix C.1) as an energy model, against a classical Hopfield network (Hopfield, 1982) using different writing rules (Storkey & Valabregue, 1999) and a recently proposed differential plasticity model (Miconi et al., 2018). It is worth noting the differentiable plasticity model is a generalized variant of Fast Weights (Ba et al., 2016), where the plasticity of each activation is modulated separately. We also consider an LSTM (Hochreiter & Schmidhuber, 1997), Memory

network (Weston et al., 2014) and a Memory-Augmented Neural Network (MANN) used by Santoro et al. (2016) which is a variant of the DNC (Graves et al., 2016).

Since the Hopfield network has limited capacity that is strongly tied to input dimensionality and that cannot be increased without adding more inputs, we use its memory size as a reference and constrain all other baseline models to use the same amount of memory. For this task it equals to $128 * (128 - 1)/2 + 128$ to parametrize a symmetric matrix and a frequency vector. We measure Hamming distance between the original and the retrieved pattern for each system, varying the number of stored patterns. We found it difficult to train the recurrent baselines on this task, so we let all models clamp non-distorted bits to their true values at retrieval which significantly stabilized training.

As we can see from the results shown in Table 1, EBMM learned a highly efficient associative memory. Only the EBMM and the memory network could achieve near-zero error when storing 64 vectors and even though EBMM could not handle 96 vectors with this number of parameters, it was the most accurate memory model.

### A.3 EVALUATION OF NON-MEMORY BASELINES

For a more complete experimental study, we additionally evaluate a number of baselines which have no capacity to adapt to patterns that otherwise would be written into the memory but still learn a prior over a domain and hence can serve as a form of associative memory.

One should note, however, that even though such models can, in principle, achieve relatively good performance by pretraining on large datasets and adopting well-designed architectures, they ultimately fail in situations where a strong memory is more important a good prior. One very distinctive example of such setting is the binary string experiment presented in the previous section, where no prior can be useful at all. EBMM, in contrast, naturally learn both the prior in the form of shared features and memory in the form of writable weights.

Our first non-memory baseline is a Variational Auto-encoder (VAE) model with 256 latent Gaussian variables and Bernoulli likelihood. VAE defines an energy equal to the negative joint log-likelihood:

$$E(\mathbf{x}, \mathbf{z}) = -\log \mathcal{N}(\mathbf{z}|0, I) - \log p(\mathbf{x}|\mathbf{z}).$$

We consider attractor dynamics similar to the one used by Wu et al. (2018). We start from a configuration $\mathbf{x}^{(0)} = \tilde{\mathbf{x}}, \mathbf{z}^{(0)} = \mu(\mathbf{x})$, where $\mu(\mathbf{x})$ is the output of a Gaussian encoder $q(\mathbf{z}|\mathbf{x}) = \mathcal{N}(\mathbf{z}|\mu(\mathbf{x}), \text{diag}(\sigma(\mathbf{x})))$. Then we alternate between updating these two parts of the configuration as follows:

$$\mathbf{z}^{(t+1)} = \mathbf{z}^{(t)} - \gamma \nabla_{\mathbf{z}} E(\mathbf{x}, \mathbf{z}^{(t)}),$$
$$\mathbf{x}^{(t+1)} = \arg\max_{\mathbf{x}} E(\mathbf{x}, \mathbf{z}^{(t+1)}).$$

Since we use a simple factorized Bernoulli likelihood model, the exact minimization with respect to $\mathbf{x}$ can be performed analytically.

One can hope that under certain circumstances a well-trained VAE would assign lower energy levels to less distorted versions of $\tilde{\mathbf{x}}$ and an image We used 50, 100 and 200 iterations for Omniglot, CIFAR and ImageNet experiments respectively, in each case also performing a grid search for the learning rate $\gamma$.

Another baseline is a Denoising Auto-encoder (DAE) which is trained to reconstruct the original pattern $\mathbf{x}$ from its distorted version $\tilde{\mathbf{x}}$ using a bottleneck of 256 hidden units. This model is trained exactly as our model using (2) as an objective and hence, in contrast to VAE, can adapt to a particular noise pattern, which makes it a stronger baseline.

Finally, we consider the Deep Image Prior (Ulyanov et al., 2018), a method that can be seen as a very special case of the energy-based model which also iteratively adapts parameters of a convolutional network to generate the most plausible reconstruction. We used the network provided by the authors and performed 2000 Adam updates for each of the images. Similarly to how it was implemented in the paper, the network was instructed about location of the occluded block which arguably is a strong advantage over all other models.

The quantitative results of the comparison are provided in the Table 2. Clearly, the most efficient of the baselines is DAE, largely because it was explicitly trained against a known distortion model. In

| Baseline | Omniglot | CIFAR | ImageNet |
|---|---|---|---|
| 25% block noise | | | |
| Variational Auto-encoder | 109.08 | 544.72 | 1889.79 |
| Denoising Auto-encoder | 36.08 | 24.66 | 141.07 |
| Deep Image Prior | – | – | 170.36* |
| 15% salt and pepper noise | | | |
| Variational Auto-encoder | 84.08 | 430.33 | 1385.29 |
| Denoising Auto-encoder | 9.57 | 8.82 | 69.20 |
| Deep Image Prior | – | – | 467.92 |

Table 2: Reconstruction errors (Hamming for Omniglot, squared for CIFAR and ImageNet) for non-memory baselines. *Location of the distorted block was used.

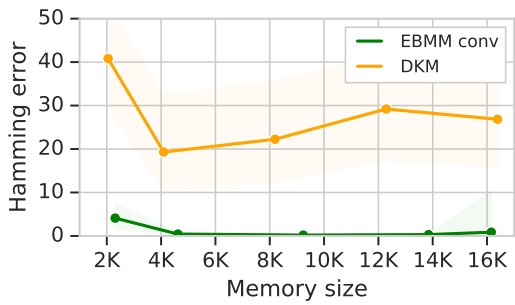

Figure 7: Reconstruction error on Omniglot. Dynamic Kanerva Machine is compared to EBMM with convolutional memory. 15% salt and pepper noise is used.

contrast, VAE failed to recover from noise producing a significantly shifted observation distribution. Deep Image Prior performed significantly better, however, since it only adapts to a single distorted image instead of a whole distribution, it could not over-perform DAE with a much simpler architecture. As expected, none of the non-memory baselines performed even comparably to models with memory we consider in the main experiments section.

## A.4 Comparison with Dynamic Kanerva Machine

As we reported earlier, Dynamic Kanerva Machine failed to perform better than a random guess under the strong block noise we used in the main paper. In this appendix we evaluate DKM in simpler conditions where it can be reasonably compared to EBMM. Thus, we trained DKM on batches of 16 images and used a relatively simple 15% salt and pepper noise.

Figure 7 contains the reconstruction error in the same format as Figure 3a. One can see that DKM generally performs poorly and does not show a consistent improvement with increasing memory size. As can be seen on Figure 8, DKM is able to retrieve patterns that are visually similar to the originally stored ones, but also introduces a fair amount of undesirable variability and occasionally converges to a spurious pattern. The errors increase with stronger block noise and larger batch sizes.

## A.5 Generalization to different distortion models

In this experiment we assess how EBMM trained with one of block noise of one size performs with different levels of noise. Figure 9 contains the generalization analysis with different kinds of noise on Omniglot.

One can see that regardless of noise type, EBMM successfully generalizes to weaker noise and generalization to slightly stronger noise is also observed. One should note though that $20 \times 20$ block noise already covers almost $40\%$ of the image and hence may completely occlude certain characters.

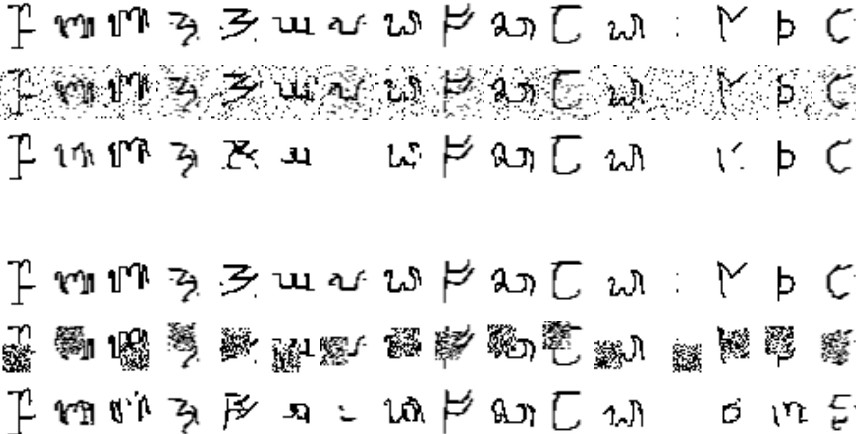

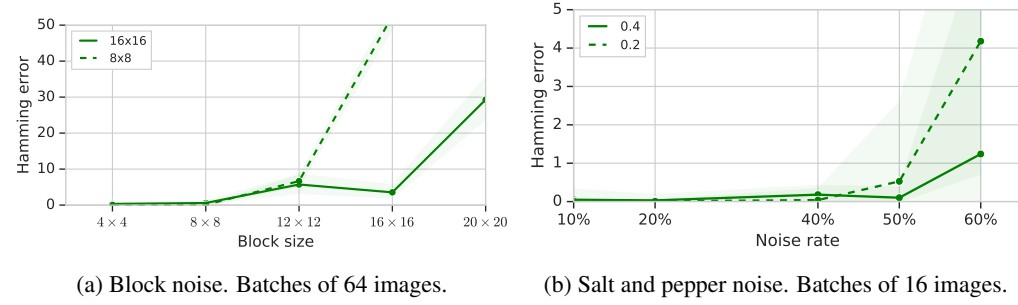

Figure 8: An example of a retrieval of a single batch by Dynamic Kanerva Machine. A model with 12K memory was used. Top: salt and pepper noise, bottom: $16 \times 16$ block noise. First line: original patterns, middle line: distorted queries, bottom line: retrieved patterns.

|  |  |
|---|---|
| (a) Block noise. Batches of 64 images. | (b) Salt and pepper noise. Batches of 16 images. |

Figure 9: Generalization to different noise intensity on Omniglot. Each line represents a model trained with a certain noise level. All models use 16K memory.

Generally we found salt and pepper noise much simpler which originally motivated us to focus on block occlusions. However, as we indicate on the figure, the model used in the experiment with this kind of noise was only trained on batches of 16 images and hence the comparison of absolute values between Figures 9a and 9b is not possible. The relative performance degradation with increasing noise rate is still representative though.

We did not observe generalization from *type* of noise to another. Perhaps, the very different kinds of distortions were not compatible with learned basins of attraction. In general, this is not very surprising as, arguably, no model can be expected to adapt to all possible distortion models without a relevant supervision of some sort.

For this experiment we did not anyhow fine-tune model parameters, including the meta-parameters of the reading gradient descent. Hence, it is possible that some amount of fine-tuning and, perhaps, a custom optimization strategy for the modelled energy may lead to an improvement.

## A.6 GENERALIZATION TO MNIST

A natural question is whether a model trained on one task or dataset strictly overfits to its features or whether it can generalize to similar, but previously unseen tasks. One of the standard experiments to test this ability is transfer from Omniglot to MNIST, since both datasets consist of handwritten characters, which, however, differ enough to present a distributional shift.

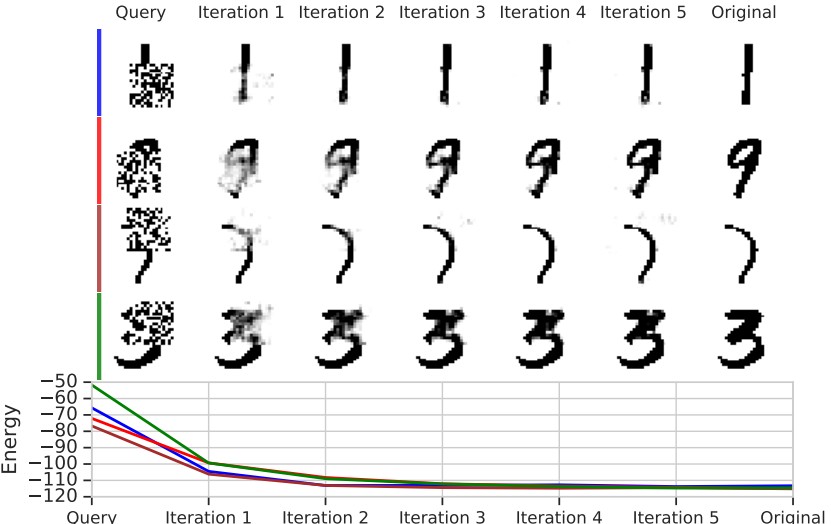

Figure 10: Iterative reading on MNIST. The model is learned on Omniglot.

| BATCH SAMPLING | 4K MEMORY | 5K MEMORY |
|---|---|---|
| UNIFORM (UNCORRELATED) | 7.18 | 5.37 |
| 2 CLASSES (CORRELATED) | 6.22 | 2.57 |

Table 3: Hamming error of models trained on differently constructed Omniglot batches of 32 images.

Developing such transfer capabilities is out of scope for this paper, but a simple check confirmed that EBMM can successfully retrieve upscaled (to match the dimensionality) and binarized MNIST characters as one can see on Figure 10. One can see that although some amount of artifacts is introduced the retrieved images are clearly recognizable and the energy levels are as adequate as in the Omniglot experiment. As in the previous experiment, we did not fine-tune the model.

### A.7 EXPERIMENTS WITH CORRELATED BATCHES

One of the desirable properties of a memory model would be an efficient consolidation of similar patterns, e.g. corresponding to the same class of images, in the interest of better compression.

We do not observe this property in the models trained on uniformly sampled, uncorrelated batches, perhaps because the model was never incentivized to do so. However, we performed a simple experiment where we trained EBMM on the Omniglot images, making sure that each batch of 32 images has characters of exactly two classes. For this purpose we employed a modified convolutional model (see Appendix C.5) in which the memory weights are located in the second residual layer instead of the third as in the main architecture. As we found experimentally, this way the model could compress more visual correlations in the batch.

The results can be found in Table 3. It is evident that training on correlated batches enabled better compression and EBMM is able to learn an efficient consolidation strategy if the meta-learning is set up appropriately.

## B READING IN EBMM

### B.1 PROJECTED GRADIENT DESCENT

We described the basic reading procedure in section 2, however, there is a number of extensions we found useful in practice.

Since in all experiments we work with data constrained to the $[0, 1]$ interval, one has to ensure that the read data also satisfies this constraint. One strategy that is often used in the literature is to model the output as an argument to a sigmoid function (logits). This may not work well for values close to the interval boundaries due to vanishing gradient, so instead we adopted a projected gradient descent, i.e.

$$\mathbf{x}^{(k+1)} = \text{proj}(\mathbf{x}^{(k)} - \gamma^{(k)} \nabla_{\mathbf{x}} E(\mathbf{x}^{(k)})),$$

where the proj function clips data to the $[0, 1]$ interval.

Quite interestingly, this formulation allows more flexible behavior of the energy function. If a stored pattern $\mathbf{x}$ has one of the dimensions exactly on the feasible interval boundary, e.g. $x_j = 0$, then $\nabla_{x_j} E(\mathbf{x})$ does not necessarily have to be zero, since $x_j$ will not be able to go beyond zero. We provide more information on the properties of storied patterns in further appendices.

### B.2 NESTEROV MOMENTUM

Another extension we found useful is to employ Nesterov momentum (Nesterov, 1983) into the optimization scheme and we use it in all our experiments.

$$\hat{\mathbf{x}}^{(k)} = \text{project}(\mathbf{x}^{(k-1)} + \psi^{(k)} v^{(k-1)}), \quad v^{(k)} = \psi^{(k)} v^{(k-1)} - \gamma^{(k)} \nabla E(\hat{\mathbf{x}}^{(k)})$$
$$\mathbf{x}^{(k)} = \text{project}(v^{(k)}).$$

### B.3 STEP SIZES

To encourage learning converging attractor dynamics we constrained step sizes $\gamma$ to be a non-increasing sequence:

$$\gamma^{(k)} = \gamma^{(k-1)} \sigma(\eta^{(k)}), \quad k > 1$$

Then the actual parameters to meta-learn is the initial step size $\gamma^{(1)}$ and the logits $\boldsymbol{\eta}$. We apply a similar parametrization to the momentum learning rates $\boldsymbol{\psi}$.

### B.4 STEP-WISE RECONSTRUCTION LOSS

As has often been found helpful in the literature (Belanger et al., 2017; Antoniou et al., 2018) we apply the reconstruction loss (2) not just to the final iterate of the gradient descent, but to all iterates simultaneously:

$$\mathcal{L}_K(X, \boldsymbol{\theta}) = \sum_{k=1}^{K} \frac{1}{N} \sum_{i=1}^{N} \mathbb{E}\left[ ||\mathbf{x}_i - \mathbf{x}_i^{(k)}||_2^2 \right].$$

## C ARCHITECTURE DETAILS

Below we provide pseudocode for computational graphs of models used in the experiments. All modules containing memory parameters are specifically named as `memory`.

### C.1 GATED RNN

We used a fairly standard recurrent architecture only equipped with an update gate as in (Chung et al., 2014). We unroll the RNN for 5 steps and compute the energy value from the last hidden state.

```
hidden_size = 1024
input_size = 128
# 128 * (128 - 1) / 2 + 128 parameters in total
dynamic_size = (input_size - 1) // 2

state = repeat_batch(zeros(hidden_size))
memory = Linear(input_size, dynamic_size)
```

```
gate = Sequential([
    Linear(input_size + hidden_size, hidden_size),
    sigmoid
])

static = Linear(input_size + hidden_size, hidden_size - dynamic_size)

for hop in xrange(5):
    z = concat(x, state)

    dynamic_part = memory(x)
    static_part = static(z)
    c = tanh(concat(dynamic_part, static_part))
    u = gate(z)
    state = u * c + (1 - u) * state

energy = Linear(1)(state)
```

## C.2  RESNET, FULLY-CONNECTED MEMORY

```
channels = 32
hidden_size = 512
representation_size = 512
static_size = representation_size - dynamic_size

state = repeat_batch(zeros(hidden_size))

encoder = Sequential([
    ResBlock(channels * 1, kernel=[3, 3], stride=2, downscale=False),
    ResBlock(channels * 2, kernel=[3, 3], stride=2, downscale=False),
    ResBlock(channels * 3, kernel=[3, 3], stride=2, downscale=False),
    flatten,
    Linear(256),
    LayerNorm()
])

gate = Sequential([
    Linear(hidden_size),
    sigmoid
])

hidden = Sequential([
    Linear(hidden_size),
    tanh
])

x = encoder(x)

memory = Linear(input_size, dynamic_size)

dynamic_part = memory(x)
static_part = Linear(static_size)(x)
x = tanh(concat(dynamic_part, static_part))

for hop in xrange(3):
    z = concat(x, state)
    c = hidden(z)
    c = LayerNorm()(c)
    u = gate(z)
```

```
    state = u * c + (1 - u) * c

h = tanh(Linear(1024)()(state))
energy = Linear(1)(h)
```

The `encoder` module is also shared with all baseline models together with its transposed version as a decoder.

## C.3 ResNet, convolutional memory

```
channels = 32
x = ResBlock(channels * 1, kernel=[3, 3], stride=2, downscale=True)(x)
x = ResBlock(channels * 2, kernel=[3, 3], stride=2, downscale=True)(x)

def resblock_bottleneck(x, channels, bottleneck_channels, downscale=False):
  static_size = channels - dynamic_size

  z = x

  x = Conv2D(bottleneck_channels, [1, 1])(x)
  x = LayerNorm()(x)
  x = tanh(x)

  if downscale:
    memory_part = Conv2D(dynamic_size, kernel=[3, 3], stride=2, downscale=True)(x)
    static_part = Conv2D(static_size, kernel=[3, 3], stride=2, downscale=True)(x)
  else:
    memory_part = Conv2D(dynamic_size, kernel=[3, 3], stride=1, downscale=False)(x)
    static_part = Conv2D(static_size, kernel=[3, 3], stride=1, downscale=False)(x)
  x = concat([static_part, memory_part], -1)
  x = LayerNorm)(x)
  x = tanh(x)

  z = Conv2D(channels, kernel=[1, 1])(z)
  if downscale:
    z = avg_pool(z, [3, 3] + [1], stride=2)
    x += z
  return x

x = resblock_bottleneck(x, channels * 4, channels * 2, False)
x = resblock_bottleneck(x, channels * 4, channels * 2, True)

recurrent = Sequential([
  Conv2D(hidden_size, kernel=[3, 3], stride=1),
  LayerNorm(),
  tanh
])

update_gate = Sequential([
  Conv2D(hidden_size, kernel=[1, 1], stride=1),
  LayerNorm(),
  sigmoid
])

hidden_size = 128
hidden_state = repeat_batch(zeros(4, 4, hidden_size))

for hop in xrange(3):
  z = concat([x, hidden_state], -1)
```

```
candidate = recurrent(z)
u = update_gate(z)
hidden_state = u * candidate + (1. - u) * hidden_state

x = Linear(1024)(x)
x = tanh(x)
energy = Linear(1)
```

## C.4 RESNET, IMAGENET

This network is effectively a slightly larger version of the ResNet with convolutional memory described above.

```
channels = 64
dynamic_size = 8

x = ResBlock(channels * 1, kernel=[3, 3], stride=2, downscale=True)(x)
x = ResBlock(channels * 2, kernel=[3, 3], stride=2, downscale=True)(x)

x = resblock_bottleneck(x, channels * 4, channels * 2, True)
x = resblock_bottleneck(x, channels * 4, channels * 2, True)

recurrent = Sequential([
  Conv2D(hidden_size, kernel=[3, 3], stride=1),
  LayerNorm(),
  tanh
])

update_gate = Sequential([
  Conv2D(hidden_size, kernel=[1, 1], stride=1),
  LayerNorm(),
  sigmoid
])

hidden_size = 256
hidden_state = repeat_batch(zeros(4, 4, hidden_size))

for hop in xrange(3):
  z = concat([x, hidden_state], -1)
  candidate = recurrent(z)
  u = update_gate(z)
  hidden_state = u * candidate + (1. - u) * hidden_state

  x = Linear(1024)(x)
  x = tanh(x)
  energy = Linear(1)
```

## C.5 RESNET, CONVOLUTIONAL LOWER-LEVEL MEMORY

This architecture is similar to other convolutional architectures with only difference is that dynamic weights are in the second residual block.

```
channels = 32
dynamic_size = 8

def resblock(x, channels):
  static_size = channels - dynamic_size

  z = x
```

```
memory_part = Conv2D(dynamic_size, kernel=[3, 3], stride=2, downscale=True)(x)
static_part = Conv2D(static_size, kernel=[3, 3], stride=2, downscale=True)(x)

x = concat([static_part, memory_part], -1)
x = LayerNorm)(x)
x = tanh(x)

memory_part = Conv2D(dynamic_size, kernel=[3, 3], stride=2, downscale=True)(x)
static_part = Conv2D(static_size, kernel=[3, 3], stride=2, downscale=True)(x)

y = concat([static_part, memory_part], -1)
y += x
y = nonlinear(LayerNorm)(y))

x += y

z = Conv2D(channels, [1, 1])(z)
z = avg_pool(z, [3, 3], 2)
x += z

return x

x = ResBlock(channels * 1, kernel=[3, 3], stride=2, downscale=True)(x)
x = resblock(x, channels * 2)
x = ResBlock(channels * 4, kernel=[3, 3], stride=2, downscale=True)(x)

recurrent = Sequential([
  Conv2D(hidden_size, kernel=[3, 3], stride=1),
  LayerNorm(),
  tanh
])

update_gate = Sequential([
  Conv2D(hidden_size, kernel=[1, 1], stride=1),
  LayerNorm(),
  sigmoid
])

hidden_size = 256
hidden_state = repeat_batch(zeros(4, 4, hidden_size))

for hop in xrange(3):
  z = concat([x, hidden_state], -1)
  candidate = recurrent(z)
  u = update_gate(z)
  hidden_state = u * candidate + (1. - u) * hidden_state

  x = Linear(1024)(x)
  x = tanh(x)
  energy = Linear(1)
```

## C.6   THE ROLE OF SKIP-CONNECTIONS IN ENERGY MODELS

Gradient-based meta-learning and EBMM in particular rely on the expressiveness of not just the forward pass of a network, but also the backward pass that is used to compute a gradient. This may require special considerations about the network architecture.

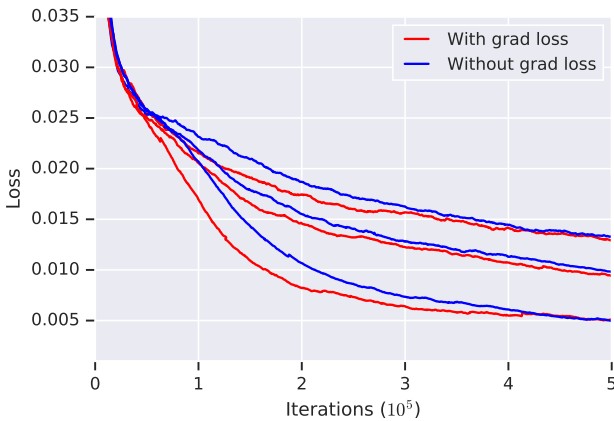

Figure 11: Effect of including the $||\nabla_{\mathbf{x}} E(\mathbf{x})||^2$ term in the writing loss (3) on Omniglot.

One may notice that all energy models considered above have an element of recurrency of some sort. While the recurrency itself is not crucial for good performance, skip-connections, of which recurrency is a special case, are.

We can illustrate this by considering an energy function of the following form:

$$E(x) = o(h(x)), \quad h(x) = f(x) + g(f(x)).$$

Here we can think of $h$ as a representation from which the energy is computed. We allow the representation to be first computed as $f(x)$ and then to be refined by adding $g(f(x))$.

During retrieval, we use gradient of the energy with respect to $x$ which can be computed as

$$\frac{d}{dx}E(x) = \frac{do}{dh}\frac{dh}{dx} = \frac{do}{dh}\left(\frac{df}{dx} + \frac{dg}{df}\frac{df}{dx}\right).$$

One can see, that with a skip-connection the model is able to *refine the gradient* together with the energy value.

A simple way of incorporating such skip-connections is via recurrent computation. We allow the model to use a gating mechanism that can modulate the refinement and prevent from unnecessary updates. We found that usually a small number of recurrent steps (3-5) is enough for good performance.

## D  EXPLANATIONS ON THE WRITING LOSS

Our setting deviates from the standard gradient-based meta-learning as described in (Finn et al., 2017). In particular, we are not using the same loss function (naturally defined by the energy function) in adaptation and inference phases. As we explain in section 3, writing loss (3) besides just the energy term also contains the gradient term and the prior term.

Even though we found it sufficient to use just the energy value as the writing loss, perhaps not surprisingly, minimizing the gradient norm appeared to help optimization especially in the early training (see Figure 11) and lead to better final results.

We use an individual learning rate per each writable layer and each of the three loss terms, initialized at $10^{-4}$ and learned together with other parameters. We used softplus function to ensure that all learning rates remain non-negative.

