# OpenReview forum: "Meta-Learning Deep Energy-Based Memory Models"
_ICLR.cc/2020/Conference — Accept (Poster)_

### Official Review · AnonReviewer3 · 2019-10-23
**Official Blind Review #3**

**Rating:** 6

**Review:**


======================================== Update after revisions ============================================

I appreciate the effort the authors have put into the revision and the rebuttal. I'm happy to increase my score and recommend acceptance based on the revised paper.

However, I have to say that some of my worries still linger. With respect to non-memory baselines, the authors have responded that the memory based models will outperform non-memory models in cases where prior structure is less important than memory and provided a demonstration of this with an extreme example, i.e. a case with no structure (random binary strings example). I understand the point being made here, but this is a rather pedantic and uninteresting example. The authors have provided another (more interesting) example in Appendix A3 and shown that the memory-based model outperforms some simpler baselines such as the DAE even in this case. But no explanation is given for this result. Why is the memory based model outperforming the DAE in this case, given that this is an example where prior *is* very important? I'm a bit worried that the DAE results may perhaps be due to a non-optimized architecture or training setup (and what exactly is the architecture used for the DAE here)? I would appreciate it if the authors could clarify these issues in the final version.

I have also spotted several typos. For the final version, please make sure to go through the paper thoroughly at least once and fix all the typos.

========================================================================================================

This paper proposes a meta-learning approach to learning fast read and write mechanisms in an energy-based model so that a given set of images can be quickly inducted into memory and retrieved from memory with noisy queries. The paper is well-written and the proposed approach seems interesting and novel enough. However, I have some concerns about the paper that need to be addressed. Here are the main issues for me:

1) I am in general not really convinced about the supposed advantages of these attractor memory models (this paper and the earlier Kanerva machine) over more standard and much simpler approaches. For example, for the problem of retrieval from noisy queries, a more standard approach would be a simple autoencoder. Note that in an autoencoder, reading (inference) is already fast. The authors might point out that writing (training) will not be fast, which is correct. However, the meta-learning phase proposed in this paper will also not be fast and perhaps the fair comparison should be between the meta-learning phase of this paper and the standard training phase of an autoencoder. Note that the autoencoder will have additional benefits. For example, with the autoencoder, one is not constrained by memory storage requirements and can make use of a much larger set of images to train the model. This allows the model to learn a richer structure in images. Moreover, with a large enough feedforward net, one can approximate arbitrarily complex dependencies in images. However, in attractor memory models, on the other hand, one necessarily restricts oneself to a particular model class that can be expressed as gradient descent dynamics in an energy landscape both during reading and writing. This seems overly restrictive to me. So, perhaps, the authors can clarify the supposed advantages of these attractor memory models a bit better. For example, I would be interested in seeing some comparative results with, say, a denoising autoencoder model.

2) Currently, the paper only uses a specific type of “block-noise” corruption. One thing that would be nice to see is some results with other noise models. I think this is important to demonstrate that the approach is general enough to handle different kinds of noise. Also, a salt-and-pepper noise will allow the authors to compare their results with the dynamic Kanerva machine (the authors note that the DKM failed to train successfully for the block-noise used here).

3) It would be good to say something about the meta-learned parameters, theta_bar, r, tau. Is there any meaningful structure in these parameters that distinguishes them from their random initial values? Is one of these parameters more important than the others? For example, what happens if you just use generic step size decay rules for gamma and eta (or perhaps no decay at all)?

**Experience Assessment:**

I have read many papers in this area.

**Review Assessment: Checking Correctness Of Derivations And Theory:**

N/A

**Review Assessment: Checking Correctness Of Experiments:**

I assessed the sensibility of the experiments.

**Review Assessment: Thoroughness In Paper Reading:**

I read the paper at least twice and used my best judgement in assessing the paper.

---

> ### Author Response · Authors · 2019-11-14
> **Reply**
>
> We’re pleased that the reviewer found the manuscript interesting, novel, and well-written. We have addressed each of the major concerns below.
>
> > 1) I am in general not really convinced about the supposed advantages of these attractor memory models (this paper and the earlier Kanerva machine) over more standard and much simpler approaches.
>
> The point of our paper is not to manifest superiority of attractor- or energy-based models in any sense. We use this framework because it naturally allows us to 1) perform associative retrieval and 2) implement fast writing that is compatible with 1).
> We agree that both denoising and variational autoencoder frameworks can implement associative retrieval in a some sense, however, only to the extent to which any prior model can do so. In the new Appendix A.3 we evaluate VAE, DAE and Deep Image Prior (which is state of the art denoising model) and find that they perform significantly worse than EBMM and other memory baselines under the relevant test conditions. Moreover, as we explain in the text, in settings where the importance of a prior is less than the importance of memory, these models will ultimately fail. A good example of such setting is the experiment with binary strings in Appendix A.2 where it is clear that since patterns are sampled from a uniform distribution, no prior would implement associative retrieval.
>
> > Note that in an autoencoder, reading (inference) is already fast. The authors might point out that writing (training) will not be fast, which is correct. However, the meta-learning phase proposed in this paper will also not be fast and perhaps the fair comparison should be between the meta-learning phase of this paper and the standard training phase of an autoencoder.
>
> It is true that there is a relatively slow meta-learning phase required by our approach. However, unlike denoising autoencoders, once our model has finished training it can read and write very quickly. Autoencoders are capable of storing and denoising data, but require a new slow and expensive training procedure each time one wishes to store new data. Depending on the intended use case one may prefer one or the other approach. For example, variants of our EBMM approach may be useful in the case that we wish to quickly and cheaply store data into a relatively volatile memory of the recent past (e.g. as when training an RL agent). Our approach is not strictly better than autoencoders. Rather, each has potential use cases that depend on downstream goals.
>
> With respect to the comment about model class it should be noted that we can take advantage of very large and arbitrarily configured networks which included inductive biases (e.g. convolutional structure) if desired.  Thus, our models can make use of large, structured, feedforward networks in the inner loop, and are thus able to learn about the rich structure in images and other complex data types.
>
> > 2) Currently, the paper only uses a specific type of “block-noise” corruption. One thing that would be nice to see is some results with other noise models.
>
> Agreed. We have now run experiments and report results with a variety of different noise models, please refer to Appendix A.5.
>
> > 3) It would be good to say something about the meta-learned parameters, theta_bar, r, tau. Is there any meaningful structure in these parameters that distinguishes them from their random initial values?
>
> These parameters have very different roles, so it is difficult to say which ones are more important. Theta_bar is the initialization for the writing process and meta-learning these is crucial, just as in any other MAML-like model. Note that \bar{\theta} is many orders of magnitude larger in size than r and \tau. Indeed, as the initial configuration for our neural network, \bar{\theta} contains rich structured information about the data domain.
> Using generic step size decay rules for \gamma and \eta works, but the stability of training greatly improves if these parameters are trained, see e.g. “Antoniou, A., Edwards, H., & Storkey, A. (2018). How to train your MAML. arXiv preprint arXiv:1810.09502”.

---

### Official Review · AnonReviewer2 · 2019-10-28
**Official Blind Review #2**

**Rating:** 6

**Review:**

Thanks for the extensive answers. I updated my rating based on the provided clarification and extra experiments.

=====================
My understanding of eq 1-5 is that the algorithm finds an energy landscape (by modifying \theta) for each dataset (task) such that in this landscape, the inputs from the distribution are reachable by truncated gradient-descent initiating at a query (distorted input with respect to some distortion model).

1- The connection to meta-learning is unclear in your experiments. Can you elaborate on that?

2- The expectation in eq 5 is over different input patterns, which I assume that a set of input patterns belong to a task. What is that you write in memory?  For each experiment, what are the different input patterns (tasks) that you have written in the memory?

3- What is the \theta that is feed to the read function at the test time?

4- Are you testing on the tasks that you already trained on?

5- How this approach generalizes to unseen (or relatively close) task?

6- Can it recover any query that is not constructed with respect to the distortion model that is trained on? or what happens if the distorted image at test times comes from a different distortion model? (image blocking, for example)

7- How many distorted samples are used for training?


8- For the chosen tasks, I am curious to see the experimental comparison to deep image prior (Ulyanov, 2018). Deep image prior would be very similar to the read operator (although the gradient descent is over the parameter of model) without having write operations when you define the energy as MSE.



Typos and writing style:
-- The expectation in eq 2 should independently show what the expectation is taken with respect to.
-- input patters -> input patterns
-- figure 3 -> Figure 3
-- section 1 -> Section 1
-- 4 random images -> four random images
-- the Figure 5b -> Figure 5b
-- Models such as (Ba et al., 2016; Miconi et al., 2018) enable ->  Models such as Ba et al. (2016) and Miconi et al. (2018) enable

**Experience Assessment:**

I have read many papers in this area.

**Review Assessment: Checking Correctness Of Derivations And Theory:**

I carefully checked the derivations and theory.

**Review Assessment: Checking Correctness Of Experiments:**

I carefully checked the experiments.

**Review Assessment: Thoroughness In Paper Reading:**

I read the paper thoroughly.

---

> ### Author Response · Authors · 2019-11-14
> **Reply**
>
> > equations (1) - (5)
>
> Yes. Another way to view our approach is as follows: we meta-learn a set of initial parameters for our neural network, \bar{\theta}. These initial parameters correspond to an initial energy landscape. Meta-learning insures that from this point in parameter space it is easy to make only a handful of gradient updates to produce a new energy landscape that effectively stores a new batch of data into memory. Once they are stored, it is possible to retrieve memories by inputting a query and then descending the energy function to retrieve an associated memory.
>
> > 1- The connection to meta-learning is unclear in your experiments. Can you elaborate on that?
>
> Meta-learning is used in our model to learn a good set of starting parameters \theta, from which it is easy to quickly write memories into a network via gradient descent.
> Another way to say this is: In the outer learning loop we learn the initial parameters \bar{\theta}, and in the inner loop we optimize the parameters to minimize the writing loss for the memories we want to store for the current batch/episode. Thus, the model learns to get good at quickly learning (or storing) new memories.
>
> > 2- The expectation in eq 5 is over different input patterns, which I assume that a set of input patterns belong to a task. What is that you write in memory?  For each experiment, what are the different input patterns (tasks) that you have written in the memory?
>
> Our explanation of this process was confusing. Thank you for pointing out the issue here. We have fixed the explanation and mathematical notation around equations 4 & 5 to make this easier to follow. We refer the reviewer to this updated section of the text for a detailed explanation.
> Briefly, we write a batch of N patterns into memory. These N patterns are sampled randomly from a larger dataset of training (or testing - during evaluation) patterns. Then we construct a reconstruction loss as a squared difference between the originally stored patterns and patterns retrieved from randomly distorted queries. This loss can now be used to compute a stochastic gradient that updates all parameters (theta_bar, r and tau).
>
> > 3- What is the \theta that is feed to the read function at the test time?
>
> The \theta used by the read function at test time is created as follows:
> We start with the parameters of the model that have been meta-learned: \bar{\theta}.
> We update these parameters using the batch of data to be stored using the writing procedure given by  eq 4: \theta^{T}
> We then test what the model can remember by querying it via the read operation.
> Crucially, the data stored via step #2 has never been seen at test time.
>
> > Are you testing on the tasks that you already trained on?
>
> We test on a set of held-out data, respecting the original train/test splits in Omniglot and ImageNet. The general task of storing data and retrieving corrupted examples is consistent during training and test.
>
> > How this approach generalizes to unseen (or relatively close) task?
>
> As with many deep learning approaches, our models can capture the underlying statistics of the kinds of data that they are trained on (e.g. the structure of natural images as in the case of ImageNet). Our model learns general initial network parameters \bar{\theta} from which it can quickly and easily store data in a compressed format.
> In the new Appendix A.6 we verify that the Omniglot model successfully transfers to MNIST data.
>
> > 6- Can it recover any query that is not constructed with respect to the distortion model that is trained on? or what happens if the distorted image at test times comes from a different distortion model? (image blocking, for example)
>
> As we show in the new Appendix A.5 - to some extent, yes. The model perfectly generalizes to smaller levels of noise and performs reasonably well with larger levels. We did not observe generalization to a different distortion model, which would indeed be a nice property, but such generalization is arguably difficult to expect with respect a distortion model completely unknown during the training.
> Note that it is straightforward to use our approach to learn more general storage procedures by training across a distribution of distortion models.
>
> > 7- How many distorted samples are used for training?
>
> We trained the model for at most 2 million iterations for all of our experiments (Appendix A). For the ImageNet experiment we trained with 32 images per iteration, so the model trained on 64M distorted images.

---

> ### Author Response · Authors · 2019-11-14
> **Reply (continued)**
>
> > 8- For the chosen tasks, I am curious to see the experimental comparison to deep image prior (Ulyanov, 2018). Deep image prior would be very similar to the read operator (although the gradient descent is over the parameter of model) without having write operations when you define the energy as MSE.
>
> We agree that this indeed is a relevant baseline and we performed a series of experiments with non-memory baselines, including Deep Image Prior. Please refer to Appendix A.3 for the quantitative study. In short, they performed strictly worse than models with memory because while a prior model can produce a plausible reconstruction it is not very helpful for the task of exact recall. In the case of Deep Image Prior it is important to note that it requires privileged information about location of the occluded area (equation 6), while we work without this assumption. Without this information, the model gets confused even by the relatively simple salt and pepper distortion.
>
> > Typos and writing style:
>
> Thank you! We have fixed all of these typos and style issues.

---

### Official Review · AnonReviewer4 · 2019-11-02
**Official Blind Review #4**

**Rating:** 6

**Review:**

*Summary:*

The authors propose to tackle the associative memory problem by recasting read/write operations to read/write by optimizing the parameters/input of an energy based model. Writing is reformulated as training a parametric energy model (EBMM) to have local minima of energy w.r.t. the parameters at memorized data points. Reading is performed by performing (projected) gradient descent on the corrupted/incomplete input to minimize the energy. To ensure the operations are fast (read and write with minimal gradient steps), the authors propose to take inspiration from modern meta-learning literature and learn initialization parameters of the energy model (and other hyperparameters for GD during read/write) from which writing is fast while ensuring reading is also fast, since the models are trained to maximize read/write performance within a constrained number of gradient steps.  Experimentally, the authors show that EBMM reading performs similar to baseline methods (but better across many memory sizes) on the standard Omniglot task. On CIFAR-10 and downsized ImageNet, they show much better L2 reconstruction error of corrupted images. They also show that the learnt energy

*Recommendation:*

I believe this is a very neat idea, and utilizes large parametric models for "smart" overfitting and compression of data for the associative memory task. The proposed meta-learning approach to training the model seems to perform well across multiple simple and challenging datasets, and therefore I would recommend accept. My current recommendation is very borderline (weak accept) because of a lack of some experimental rigour (which I would love clarifications on), and missing related work, which I mention below.

*Discussion Points and Concerns from the Reviewer:*

- Dataset / batching details
Please mention how the datasets were split for training and testing the models. How much training data is utilized to meta-learn the EBMM initialization? How is batching performed? I believe these details are very important to mention in the paper for reproducibility of results.

Are there any correlations in the batch selection? Can you evaluate how good the associative memory performs across different correlation levels in the batch (A well learnt algorithm should demonstrate better reconstruction at lower memory levels for correlated batches).

- Experiments across multiple SNR and generalization on noise patterns
The authors mention at the beginning of Section 4 that a random block is corrupted, but in the end the experiments are done on a constant corruption size on the CIFAR and ImageNet images. How do the models perform across different signal-to-noise ratios? Similarly, the model is trained on simple noise patterns

- Missing related work
There is related work [1] in learning in Hopfield Networks using the implicit function theorem and finding stationary points of the dynamics. This work is not mentioned in the paper, and is a valid baseline for this paper as well.

- Mentioning Appendix D in the main paper
Appendix D is not mentioned in the main paper and has a short discussion on the mismatch between the reading process and the writing loss during meta-training. It also mentions additional tricks required for the training, and I believe it should be mentioned in the main paper like other sections are appropriately referenced.

- Large batch sizes for ImageNet
Work from [2] can be utilized to backpropagate through very long optimization sequences and therefore can be utilized to train with larger batch sizes in the ImageNet example. It is important to see how the small model utilized for ImageNet works to compress higher batch sizes, as that is one of the major practical issues with the algorithm.

- Related paper at NeurIPS this year
[3] is a related paper from Neurips this year, which the authors could consider adding as contemporary work

- Comments on scalability
The associative memory papers have often been criticized for lack of scalability, and I think the authors make progress towards making this better with the use of unconstrained energy models in the learning process. It would be nice to have a discussion of the scalability from the authors, highlighting issues in the current model and future directions

References:
[1] Reviving and Improving Recurrent Back-Propagation, ICML '18
[2] Gradient-based Hyperparameter Optimization through Reversible Learning, Maclaurin et al. ICML '15
[3] Metalearned Neural Memory, Munkhdalai et al. NeurIPS '19

**Experience Assessment:**

I have read many papers in this area.

**Review Assessment: Checking Correctness Of Derivations And Theory:**

I carefully checked the derivations and theory.

**Review Assessment: Checking Correctness Of Experiments:**

I carefully checked the experiments.

**Review Assessment: Thoroughness In Paper Reading:**

I read the paper at least twice and used my best judgement in assessing the paper.

---

> ### Author Response · Authors · 2019-11-14
> **Reply**
>
> > - Dataset / batching details
>
> For Omniglot, Cifar and ImageNet, we used the natural train/test splits in the datasets. We perform meta-learning training for at most 2 million iterations. For Omniglot and Cifar we used a batch size of 64, for ImageNet we select a batch of 32 images at a time. Batches were sampled uniformly. We have made this much clearer in the text now.
>
> > - Experiments across multiple SNR and generalization on noise patterns
>
> Thank you for the suggestion, we have now added Appendix A.5 where we test generalization to different SNR on Omniglot. The model perfectly adapts to lower SNR and is able to perform reasonably well with non-significantly stronger noise.
> We will be able to provide a more complete study in the final version of the paper.
>
> > - Missing related work
>
> This is indeed a relevant paper, thank you for the suggestion. Training attractor models using implicit differentiation is a promising direction for future work.
>
> > - Large batch sizes for ImageNet
>
> This work is very relevant, we now cite it.
>
> > - Mentioning Appendix D in the main paper
>
> The appendix is now referenced.
>
> > - Related paper at NeurIPS this year
>
> We agree it is a very relevant paper and we already cite it in the Related work section. To avoid any confusion, we would like to emphasize that our works differ in the interface of a memory module. In the MNM it is the key-value retrieval of non-structured vectors, while EBMM is focused on more association problems, e.g. where any part of the stored pattern can be used as a key.
>
> > - Comments on scalability
>
> Scalability can be understood as a multi-dimensional concept. We made an improvement on the expressivity (by adopting modern deep learning techniques) and the speed (by utilizing gradient-based meta-learning) dimensions, but we do not demonstrate yet an advantage over slot-based memory in temporal tasks with incremental updates. Although our early experiments suggest that EBMM is more than viable in this setting, training on long sequences is less straightforward than traditional recurrent models. The papers you suggested as related work can be very helpful to improve on this dimension. We have refined Section 6 to reflect on this.

---

> ### Author Response · Authors · 2019-11-15
> **Reply (continued)**
>
> > Experiments on correlated batches
>
> We display results for randomly chosen images within the test set. If storing very similar images this can actually make correct reconstruction more difficult, as there is more ambiguity in locating the original image from the occluded query image. We found that if the model was not trained on correlated batches, it does not benefit from them at the test time. However, when trained and tested on batches of 2 Omniglot classes, EBMM achieves significantly lower reconstruction error. Please see the new Appendix A.7 for details. We will be able to provide a comprehensive study for the final version of the paper.

---

### Official Review · AnonReviewer1 · 2019-11-04
**Official Blind Review #1**

**Rating:** 6

**Review:**

Summary:
This paper proposes a new type of energy-based models, a class of non-normalized generative models that relies on an energy function to retrieve patterns that correspond to its minima. The goal that is tackled by the authors is to implement an associative memory system, i.e. a mechanism that is able to retrieve any one of a set of patterns, given a distorted copy of these patterns. This task is traditionally carried out using attractor neural networks like the Hopfield model, a recurrent neural network model endowed with a learning rule that allows it to quickly embed a given set of patterns in its weight matrix such that the patterns become stable fix points of its dynamics. As the authors point out though, models like the Hopfield model are limited in their capacity to assimilate attractor patterns and in term of their expressiveness. On the other hand, more complex models based on deep architectures trained with gradient descent are slow at updating their weights to create new attractors.
The authors propose a new method to make up for the weaknesses of these two approaches. Their method is based on meta-learning, and in short consists in meta-training an energy function parametrized as a neural network such that  executing a write dynamics on the weights results in a model whose read dynamics (a gradient descent on the energy function) is able to denoise distorted inputs and retrieve the original ones. In practice, the write dynamics is obtained as a gradient descent procedure on a writing loss (which is itself dependent on the energy function) as a function of the weights. The meta-learning procedure minimizes the discrepancy between the original patterns and the retrieved ones by optimizing end-to-end the learning schedule parameters and initial conditions of the weights, analogously to gradient-based meta-learning methods like MAML.
The authors then carry out a series of experiments to check that their model is competitive with Memory-Augmented Neural Network (MANN) and Memory Networks (MemNets) in retrieving samples from Omniglot, CIFAR and ImageNet, in terms of retrieving abilities for a given memory size. In the supplementary material section they in addition compare their model's performance against the Hopfield model and recurrent networks on the classical toy task of retrieving random binary patterns, also with good results for the new model.

Decision:
This paper is very clearly and compactly written. The idea of training an energy-based model through gradient-based meta-learning seems novel and innovative.
One thing that the the paper is arguably missing, is a convincing motivation section for focusing on energy-based models. The panorama of generative models has radically changed since attractor neural networks and energy-based models were first introduced. At the time powerful methods like variational autoencoders, normalizing flows and GANs didn't exist. But nowadays, one could arguably expect that energy-based models should be contextualized and motivated in the perspective of comparing them with these new breeds of deep generative models. I am absolutely not suggesting that the authors should providing experimental comparisons between their models and GAN or VAE, but simply that they compare them to their style of generative modeling in terms of advantages, disadvantages, use cases, and potential for applications.

**Experience Assessment:**

I have published in this field for several years.

**Review Assessment: Checking Correctness Of Derivations And Theory:**

I carefully checked the derivations and theory.

**Review Assessment: Checking Correctness Of Experiments:**

I assessed the sensibility of the experiments.

**Review Assessment: Thoroughness In Paper Reading:**

I read the paper thoroughly.

---

> ### Author Response · Authors · 2019-11-14
> **Reply**
>
> Thank you for your review. We agree that a discussion about our model and modern models such as GANs and VAEs was somewhat missing in our initial submission. We have added a paragraph in the Related work section to better position EBMM within modern deep learning and also performed a comparison with a number of non-memory baselines (Appendix A.3). We hope this confirms both conceptual and empirical contributions of our paper.

---

### Author Response · Authors · 2019-11-14
**From authors**

We would like to thank our reviewers for their time and valuable feedback, many of the comments helped us to improve the paper and obtain more results. We will soon be replying directly to each of the reviewers. Some of the requested experiments are still running and we will be updating the paper with new results.
We believe we positively addressed most of the feedback and ask the reviewers to assess our replies.

---

### Author Response · Authors · 2019-11-15
**Summary of changes**

We would like to summarize the changes we made to the initial submission based on the reviewers feedback.

1) The training process is clarified.
2) A discussion of EBMM in the context of other modern techniques such as GANs and VAEs is added to Section 5.
3) Section 6 has been enriched with a more detailed discussion of existing limitations of EBMM and potential ways of improving on them.
4) Appendix A.3 has been added where we provide a direct comparison to three different non-memory baselines.
5) Comparison to Kanerva Machine has been added as Appendix A.4
6) We performed experiments with different distortion models in Appendix A.5. EBMM shows satisfactory behavior with both decreased and increased level of noise.
7) We tested the ability to transfer writing and reading capabilities of EBMM on the example of Omniglot and MNIST in Appendix A.6.
8) We performed an experiment with the correlated batches in Appendix A.7 and found that training on correlated batches leads to better memory consolidation.

We hope our reviewers and area chairs will find these changes as a significant improvement of the paper and as a comprehensive answer to all raised questions.

---

### Public Comment · ~Jianwen_Xie1 · 2020-01-02
**Missing related reference about Energy-based models using  neural networks to approximate the energy function**

Dear Authors,

Congratulations on your nice accepted paper.

I would like to point out some papers that are highly related to your current one, and hope you can cite them in your final version.  All of them are about generative models, which are in the forms of energy-based models parameterized by neural nets.

The seminal paper that proposes an energy-based model parameterized by modern deep neural network and learned it by Langevin based MLE is in (Xie. ICML 2016) [1].  The paper also involves theory about the connection with the discriminative ConvNet, Hopfield network, and Contrastive divergence.  The model is called "Energy-based" generative ConvNet, because it is naturally derived from the discriminative ConvNet, instead of manually designed.

(Xie. CVPR 2017) [2] proposed to use Spatial-Temporal ConvNet as the energy function for video modeling. In the theory part, it firstly provides a self-adversarial interpretation for the MCMC-based learning of the EBM with ConvNet as energy functions.

(Xie. CVPR 2018) [3] proposed to use volumetric ConvNet as the energy function for 3D shape patterns generation. It is called 3D descriptor Net.

(Gao. CVPR 2018) [4] proposed multi-grid MCMC to learn EBM with ConvNet as energy function.

(Nijkamp 2019) [5] proposed short-run MCMC to learn EBM with ConvNet as energy function.

Thank you :)

Reference
[1] A Theory of Generative ConvNet.
Jianwen Xie *, Yang Lu *, Song-Chun Zhu, Ying Nian Wu (ICML 2016)

[2] Synthesizing Dynamic Pattern by Spatial-Temporal Generative ConvNet
Jianwen Xie, Song-Chun Zhu, Ying Nian Wu (CVPR 2017)

[3] Learning Descriptor Networks for 3D Shape Synthesis and Analysis
Jianwen Xie *, Zilong Zheng *, Ruiqi Gao, Wenguan Wang, Song-Chun Zhu, Ying Nian Wu (CVPR) 2018

[4]  Learning generative ConvNets via multigrid modeling and sampling.
R Gao*, Y Lu*, J Zhou, SC Zhu, and YN Wu (CVPR 2018).

[5] On learning non-convergent non-persistent short-run MCMC toward energy-based model.
E Nijkamp, M Hill, SC Zhu, and YN Wu (NeurIPS 2019)

---

### Decision · Program_Chairs · 2019-12-19

**Decision:**

Accept (Poster)

**Comment:**

Four knowledgable reviewers recommend accept. Good job!